# Total Synthesis and Anti-Inflammatory Bioactivity of (−)-Majusculoic Acid and Its Derivatives

**DOI:** 10.3390/md19060288

**Published:** 2021-05-21

**Authors:** Hong-Xiu Xiao, Qing-Xiang Yan, Zhi-Hui He, Zheng-Biao Zou, Qing-Qing Le, Ting-Ting Chen, Bing Cai, Xian-Wen Yang, Su-Lan Luo

**Affiliations:** 1Key Laboratory of Tropical Biological Resources of Ministry of Education, Hainan University, Haikou 570228, China; xiaohongxiu97@hainanu.edu.cn; 2Key Laboratory of Marine Biogenetic Resources, Third Institute of Oceanography, Ministry of Natural Resources, 184 Daxue Road, Xiamen 361005, China; youngqx@126.com (Q.-X.Y.); hezhihui@tio.org.cn (Z.-H.H.); zhengbiaozou@njust.edu.cn (Z.-B.Z.); leqingqing@tio.org.cn (Q.-Q.L.); chentingting@tio.org.cn (T.-T.C.); caibing@tio.org.cn (B.C.)

**Keywords:** marine natural products, (−)-majusculoic acid, anti-inflammation, LPS, CCK-8

## Abstract

The first total synthesis of marine natural product, (−)-majusculoic acid (**1**) and its seven analogs (**9**–**15**), was accomplished in three to ten steps with a yield of 3% to 28%. The strategy featured the application of the conformational controlled establishment of the *trans*-cyclopropane and stereochemical controlled bromo-olefination or olefination by Horner–Wadsworth–Emmons (HWE) reaction. The potential anti-inflammatory activity of the eight compounds (**1** and **9**–**15**) was evaluated by determining the nitric oxide (NO) production in the lipopolysaccharide (LPS)-induced mouse macrophages RAW264.7. (−)-Majusculoic acid (**1**), methyl majusculoate (**9**), and (1*R*,2*R*)-2-((3*E*,5*Z*)-6-bromonona-3,5-dien-1-yl)cyclopropane-1-carboxylic acid (**12**) showed significant effect with inhibition rates of 33.68%, 35.75%, and 43.01%, respectively. Moreover, they did not show cytotoxicity against RAW264.7 cells, indicating that they might be potential anti-inflammatory agents.

## 1. Introduction

Inflammation can be triggered by tissue injury, infection, and malfunction. Chronic inflammatory conditions are related to a wide range of diseases, such as atherosclerosis and type II diabetes [1]. Lipopolysaccharide (LPS) is one of the main components of the Gram-negative bacteria membrane, which could be recognized by toll-like receptors and then result in inflammatory reaction [2]. LPS-mediated activation of toll-like receptors can activate the downstream nuclear transcription factor κB (NF-κB), activator protein 1 (AP-1), and mitogen-activated protein kinase signaling pathway (MAPK) [3]. Inducible nitric oxide synthase (iNOS) plays an important role in up-regulating NO levels [4]. Overexpression of iNOS can facilitate the production of NO. When immune cells are stimulated by microbial endotoxins and inflammatory mediators, etc., they will generate a large amount of iNOS to generate NO for immune response. Therefore, inhibition of NO production is a direct indicator of the anti-inflammatory activity.

Fatty acids (FAs) were reported to be effective in the suppression of inflammatory mediator production, including halogenated FAs and polyunsaturated FAs [5,6]. Those FAs could inhibit pathophysiological mediator production such as TNF-*α*, NO, and IL-6 in LPS-activated RAW 264.7 macrophages [7]. Therefore, FAs are regarded as lead compounds in the drug research and development, especially for anti-inflammatory medicines [8].

Cyclopropane fatty acids (CFAs, Figure 1) are commonly found in terrestrial plants and marine organisms including sponges and bacteria [9,10,11,12]. Since a cyclopropane motif can provide specific steric, stereo-electronic, and electronic properties [9], many CFAs showed intriguing bioactivities, such as anticancer [13], antimicrobial [11,12], antitrypanosomal [14], β-arrestin-biased agonistand [12], and topoisomerase I inhibitory activity [15].

Majusculoic acid (**1**), a marine-derived CFA with a halogen atom substituted conjugated diene, was first isolated from the community of cyanobacterial mat microbial in 2005 [11]. Its enantiomer, (+)-majusculoic acid, was total synthesized in 2018 using the dimerization–cyclopropanation–dedimerization strategy in 13 steps [16]. In this study, a more concise strategy was designed to construct a series of FAs with a *trans*-cyclopropane motif. Herein, we report total synthesis and anti-inflammatory activity of (−)-majusculoic acid and its analogs (Figure 2).

## 2. Results

### 2.1. Total Synthesis of Majusculoic Acid

As shown in Scheme 1, the aldehyde **24** was obtained through eight steps. The chiral epoxide **16**, after the nucleophilic addition reaction with the allyl magnesium bromide, provided the chlorinated secondary alcohol **17**, which was transformed into the new epoxide **18** by the treatment with NaOH [17]. By conformationally controlled establishment of the *trans*-4,6-dialkylcyclopropane through Horner reagent in the presence of NaH [18], the ester **19** was treated with isopropylmagnesium chloride and *N*,*O*-dimethylhydroxylamine hydrochloride to afford the Weinreb amide **20**. After reduction by diisobutylaluminium hydride (DIBAL-H), the amide **20** was transformed to aldehyde. Wittig reagent was used to extend the length of the chain from aldehyde **21** to alkene ester **22**. Before the ozonolytic cleavage of the double bond, the alkene ester **22** was selectively reduced by magnesium in MeOH. After the aldehyde **24** was synthesized, HWE reaction was applied to establish the conjugated diene motif. It is concise to construct the *trans*-cyclopropane unit and extend the chain.

According to the reported strategy [16], the HWE reaction was applied to construct the conjugated (*E*,*Z*)-bromodiene moiety. And (*E*,*Z*)-bromodiene **9** was obtained by the treatment of Horner reagent and NaHMDS at −78 °C. Compound **9** was hydrolyzed by LiOH to give (−)-majusculoic acid (**1**) in two steps with a yield of 18% (Scheme 2). The synthesized (−)-majusculoic acid (**1**) has the same spectroscopic properties as that of the natural counterpart [11].

### 2.2. Total Synthesis of Majusculoic Acid Derivatives

By the same strategy for the total synthesis of majusculoic acid, the construction of the conjugated (*E*,*E*)-diene element in **10** and **11** can also be realized (Scheme 3).

To investigate the effect between the length of carbon chain and the anti-inflammatory activity, compound **19** was transformed into aldehyde **25** by ozonolytic cleavage of the double bound. The homolog of majusculoic acid was realized after the HWE reaction (Scheme 4).

For the transformation of **9** to **14**, the direct opening of cyclopropane by the treatment of PtO_2_ or Pd/C under H_2_ was failed. Alternatively, **14** was constructed by the reaction between aldehyde **27** and Horner reagent. Aldehyde **27** was synthesized with methyl oleate **26**, by the ozonolytic cleavage of the double bound (Scheme 5).

### 2.3. Synthesis of Horner Reagent

#### 2.3.1. Synthesis of Horner Reagent for Bromo-Olefination (**S4** and **S5**)

The construction of Horner reagent for bromo-olefination was conducted based on the reported strategy (Scheme 6) [16].

#### 2.3.2. Synthesis of Horner Reagent (**S8**)

The construction of Horner reagent was commenced with (*E*)-hex-2-en-1-ol (**S6**), by the treatment of PBr_3_ and trimethyl phosphate (Scheme 7).

### 2.4. Anti-Inflammatory Effect of Majusculoic Acie and Its Analogs

The anti-inflammatory activity of majusculoic acid and its analogs was determined in the LPS-induced RAW264.7 macrophages. (−)-Majusculoic acid (**1**), methyl majusculoate (**9**) and ethyl-(1*R*,2*R*)-2-((3*E*,5*Z*)-6-bromonona-3,5-dien-1-yl)cyclopropane-1-carboxylate (**12**) showed weak inhibitory activity. Noteworthy, all three compounds showed dose-dependent effects (Table 1).

### 2.5. Cytotoxicity of Majusculoic Acid and Its Analogs

To investigate their cytotoxic effects, all eight compounds were tested for the cell proliferation by Cell Counting Kit 8 (CCK-8) assay. As shown in Table 2, they did not show cytotoxicity even under the concentration of 30 µM.

## 3. Discussion

Inflammation occurs when organisms receive stimuli, and manifests as pain, redness, heat, swelling, and dysfunction [19,20]. Monocytes can differentiate into macrophages after the stimulation at the site of infection, followed by the release of inflammatory mediators including prostaglandin E-2 (PGE-2), NO, and other cytokines. Inhibiting the production of those cytokines can control the inflammatory response. Therefore, finding out efficient molecules that could down-regulate the levels of those factors is a way to find potential drugs.

FAs are reported to be effective in inflammatory inhibition activity [5,6,7]. Yet the anti-inflammatory activity of the marine-derived CFAs that have a cyclopropane motif, such as majusculoic acid, remains unknown. Therefore, an efficient way to realize the total synthesis of majusculoic acid and its derivatives, as well as a primary structure–activity relationship (SAR) study on anti-inflammatory effect were figured out. A conformational controlled strategy was applied in the building of *trans*-4,6-dialkylcyclopropane and an efficient homologation strategy was involved. To construct the conjugated (*E*,*E*)-diene element and (*E*,*Z*)-bromodiene moiety stereospecificly, we developed three Horner reagents for different substrates. These derivatives were designed to identify the impact of halogen atom, length of chain, esterification as well as the unique *trans*-4,6-dialkylcyclopropane on the influence of anti-inflammatory activity.

As a result, an efficient synthetic strategy was developed to totally synthesize majusculoic acid and its derivatives with yields over 3%. Majusculoic acid (**1**), methyl majusculoate (**9**) and ethyl-(1*R*,2*R*)-2-((3*E*,5*Z*)-6-bromonona-3,5-dien-1-yl)-cyclopropane-1-carboxylate (**12**) showed weak effects on the NO production. As shown in Figure 3, the primary SAR result indicated that the existence of the *trans*-4,6-dialkylcyclopropane and (*E*,*Z*)-bromodiene are crucial for the activity and the length of the chain is key to the effect. However, the methyl esterification will not influence its bioactivity.

## 4. Materials and Methods

### 4.1. Reagents and Materials

Unless otherwise stated, all reactions were conducted under an argon atmosphere and anhydrous conditions in the dry organic solvent. Super-dry MeOH was purchased from Innochem Science & Technology Co., Ltd. (Beijing, China). The synthesized products were monitored by thin layer chromatography (TLC), and visualized by KMnO_4_ or ultra-violet (UV) lights. The high-resolution electron spray ionization mass spectra (HRESIMS) were obtained by Micromass Quadrupole/Time-of-Flight (Q-TOF) mass spectrometer (Waters Corporation, Milford, MA, USA). The nuclear magnetic resonance (NMR) spectra were measured on CDCl_3_ (*δ*_H_ = 7.26 and *δ*_C_ = 77.0) by Bruker AV-400 spectrometer (Bruker, Fällanden, Switzerland).

The RAW264.7 cells were bought from Shanghai Kanglang Biological Technology Co., Ltd. (Beijing, China). The Griess reagent kit and the CCK-8 kit were bought from the Thermo Fisher Scientific (Shanghai, China) and MedChemExpress (Shanghai, China), respectively.

### 4.2. Total Synthesis of Majusculoic Acid and Its Derivates

#### 4.2.1. Synthesis of (*S*)-2-(but-3-en-1-yl)oxirane (**18**)

To a solution of **16** (10.0 g, 108.0 mmol) in THF was added CuI (2.06 g, 10.8 mmol) and the solution was stirred for 20 min. Then Grinard reagent allyl magnesium bromide was added (118.9 mmol, 2 mol/L in THF) dropwise. The reaction mixture was warmed to 23 °C slowly and quenched by saturated NH_4_Cl solution (60 mL). The mixture was extracted with EtOAc. The combined organic layer was washed with brine, dried over MgSO_4_ and concentrated in vacuo to give a crude (*S*)-1-chlorohex-5-en-2-ol (**17**) (16.1 g), to which NaOH (8.64 g, 216 mmol) was added at 0 °C. After stirring for 10 h, NaOH was removed by filtration and the mixture was washed by diethyl ether. The combined organic layer was washed with brine, dried over MgSO_4_ and concentrated in vacuo to afford crude epoxide (**18**) (6.2 g) as yellow oil.

#### 4.2.2. Synthesis of Ethyl (1*R*,2*R*)-2-(but-3-en-1-yl) cyclopropane-1-carboxylate (**19**)

To a solution of NaH in toluene (100 mL) at 0 °C was added (EtO)_2_POCH_2_COOEt (35.5 g, 158.2 mmol) dropwise. The mixture was stirred for 30 min before adding the crude epoxide **18** (6.2 g, 63.3 mmol) in toluene (15 mL). The mixture was heated at 110 °C and stirred for 6 h. Then it was quenched with saturated NH_4_Cl solution (60 mL), followed by extraction with diethyl ether. The combined organic layer was washed with brine, dried over MgSO_4_ and concentrated in vacuo. Purification of the residue by flash chromatography (PE/EtOAc = 40:1) afforded **19** (11.36 g, 63% yield of three steps) as colorless oil.

#### 4.2.3. Synthesis of (1*R*,2*R*)-2-(but-3-en-1-yl)-*N*-methoxy-*N*-methylcyclopropane-1-carboxamide (**20**)

To a solution of ester **19** (2.8 g, 16.7 mmol) and NH(OMe)Me•HCl (1.7 g, 18.0 mmol) in THF (100 mL) at 0 °C was added *i*PrMgCl (33 mL, 66 mmol, 2.0 mol/L in THF) dropwise. The reaction mixture was slowly warmed to 23 °C and then stirred for 4 h before adding saturated NH_4_Cl solution (60 mL) dropwise. The mixture was extracted with EtOAc. The combined organic layer was washed with brine, dried over MgSO_4_ and concentrated in vacuo. The crude amide **20** (3.1 g, 16.7 mmol) was used for next step without further purification.

#### 4.2.4. Synthesis of (1*R*,2*R*)-2-(but-3-en-1-yl) cyclopropane-1-carbaldehyde (**21**)

To a solution of crude amide **20** (3.1 g, 16.7 mmol) in CH_2_Cl_2_ (5 mL) at −78 °C was added DIBAL-H (27.8 mL, 41.6 mmol, 1.5 mol/L in Et_2_O) dropwise. The mixture was stirred for 2 h before the addition of HCl (45 mL, 1 mol/L) dropwise until the solution turned to be clear. The mixture was extracted with EtOAc. The combined organic layer was washed with brine, dried over MgSO_4_ and concentrated in vacuo to afford crude aldehyde **21** (2.1 g).

#### 4.2.5. Synthesis of Methyl (*E*)-3-((1*R*,2*R*)-2-(but-3-en-1-yl) cyclopropyl) Acrylate (**22**)

To a solution of crude aldehyde **21** (2.1 g, 16.7 mmol) in CH_2_Cl_2_ (50 mL) was added Wittig reagent (13.9 g, 41.5 mmol). The mixture was stirred overnight. Then silica gel was added, and the mixture was concentrated in vacuo. Purification of the residue by flash chromatography (PE/EtOAc = 80:1) afforded **22** (1.9 g, 65% yield in three steps).

#### 4.2.6. Synthesis of Methyl 3-((1*R*,2*R*)-2-(but-3-en-1-yl) cyclopropyl) Propanoate (**23**)

To a solution of **22** (1.0 g, 5.56mmol) in MeOH (50 mL) at 0 °C was added Mg (667.0 mg, 27.8 mmol). The mixture was stirred for 1 h before adding 1 mol/L HCl (60 mL). The mixture was extracted with EtOAc. The EtOAc was combined and washed with brine, dried over MgSO_4_ and concentrated in vacuo. Purification of the residue by flash chromatography (PE/EtOAc = 100:1) afforded **23** (592 mg, 58% yield).

#### 4.2.7. Synthesis of Methyl 3-((1*R*,2*R*)-2-(3-oxopropyl) cyclopropyl) Propanoate (**24**)

Ozone was bubbled into a solution of **23** in CH_2_Cl_2_ at −78 °C until the solution turned to blue and the O_2_ was bubbled into the blue solution at −78 °C until the disappearance of the blue color. PPh_3_ (476 mg, 1.8 mmol) was added, and the resulting solution was slowly warmed to room temperature. Then it was stirred overnight. The crude mixture was combined with silica gel and concentrated in vacuo. Purification of the residue by flash chromatography (PE/EtOAc = 20:1) afforded **24** (62.2 mg, 75% yield) as a colorless oil.

#### 4.2.8. Synthesis of Methyl Majusculoate (**9**)

To a solution of **S5** [16] (415.5 mg, 1.53 mmol) in THF (20 mL) at −78 °C was added NaHMDS (0.76 mL, 2 mmol/L in THF). The mixture was stirred at −78 °C for 30 min before **24** (113.0 mg, 0.61 mmol) in THF (5 mL) was added. The mixture was stirred at −78 °C for 2 h before the addition of saturated NH_4_Cl solution (10 mL). Extraction with EtOAc and the combination of organic layer was washed with brine, dried over MgSO_4_ and concentrated in vacuo. Purification of the residue by flash chromatography (PE/EtOAc = 100:1) afforded **9** (40.4 mg, 20% yield).

#### 4.2.9. Synthesis of Majusculoic Acid (**1**)

To a solution of **9** (40.4 mg, 0.12 mmol) in THF (10 mL) and H_2_O (3 mL) was added LiOH (5.9 mg, 0.24 mmol). The mixture was stirred for 5 h. After the pH of the mixture was adjusted to about 5 with HCl (2 mol/L), it was extracted with EtOAc. The combination of organic layer was washed with brine, dried over MgSO_4_ and concentrated in vacuo. Purification of the residue by flash chromatography (PE/EtOAc = 6:1) afforded **1** (35.2 mg, 93% yield).

### 4.3. General Method for the Synthesis of Majusculoic Acid Analogs

#### 4.3.1. Synthesis of Methyl 3-((1*R*,2*R*)-2-((3*E*,5*E*)-nona-3,5-dien-1-yl) cyclopropyl) Propanoate (**10**)

To a solution of **S8** [21] (101.9 mg, 0.53 mmol) in THF (20 mL) at −78 °C was added KHMDS (0.76 mL, 2 mmol/L in THF). The mixture was stirred at −78 °C for 30 min before **24** (39.4 mg, 0.21 mmol) in THF (10 mL) was added. The mixture was stirred at −78 °C for 1.5 h before adding the saturated NH_4_Cl solution (10 mL). It was extracted with EtOAc and the combination of organic layer was washed with brine, dried over MgSO_4_ and concentrated in vacuo. Purification of the residue by flash chromatography (PE/EtOAc = 100:1) afforded **6** (26.3 mg, 50% yield)

#### 4.3.2. Synthesis of Methyl 3-((1*R*,2*R*)-2-((3*E*,5*E*)-nona-3,5-dien-1-yl) cyclopropyl) Propanoic Acid (**11**)

To a solution of **6** (26.3 mg, 0.11 mmol) in THF (10 mL) and H_2_O (3 mL) was added LiOH (5.7 mg, 0.24 mmol). The mixture was stirred for 5 h. The pH of the mixture was adjusted to about 5 with HCl (2 mol/L) and then extraction with EtOAc was conducted. The combination of organic layer was washed with brine, dried over MgSO_4_ and concentrated in vacuo. Purification of the residue by flash chromatography (PE/EtOAc = 6:1) afforded **11** (21.2 mg, 82% yield) as a colorless oil.

#### 4.3.3. Synthesis of Ethyl (1*R*,2*R*)-2-(3-oxopropyl) cyclopropane-1-carboxylate (**25**)

Ozone was bubbled into a solution of **19** in CH_2_Cl_2_ at −78 °C until the solution turned to blue and the O_2_ was bubbled into the blue solution at −78 °C until the disappearance of the blue color. PPh_3_ (1.2 g, 4.6 mmol) was added, and the resulting solution was slowly warmed to room temperature. Then it was stirred overnight. The crude mixture was combined with silica gel and concentrated in vacuo. Purification of the residue by flash chromatography (PE/EtOAc = 20:1) afforded **25** (179.2 mg, 91% yield) as a colorless oil.

#### 4.3.4. Synthesis of Ethyl (1*R*,2*R*)-2-((3*E*,5*Z*)-6-bromonona-3,5-dien-1-yl) cyclopropane-1-carboxylate (**12**)

To a solution of **S4** (143.7 mg, 0.53 mmol) in THF (20 mL) at −78 °C was added NaHMDS (0.27 mL, 2 mmol/L in THF). The mixture was stirred at −78 °C for 30 min before **25** (36.5 mg, 0.21 mmol) in THF (10 mL) was added. The mixture was stirred at −78 °C for 1.5 h before adding the saturated NH_4_Cl solution (10 mL). Extraction with EtOAc was performed and the combination of organic layer was washed with brine, dried over MgSO_4_ and concentrated in vacuo. Purification of the residue by flash chromatography (PE/EtOAc = 100:1) afforded **12** (27.8 mg, 42% yield)

#### 4.3.5. Synthesis of (1*R*,2*R*)-2-((3*E*,5*Z*)-6-bromonona-3,5-dien-1-yl) cyclopropane-1-carboxylic Acid (**13**)

To a solution of **12** (27.8 mg, 0.09 mmol) in THF (10 mL) and H_2_O (3 mL) was added LiOH (4.32 mg, 0.18 mmol). The mixture was stirred for 5 h. The pH of the mixture was adjusted to about 5 with HCl (2 mol/L) and then extraction with EtOAc was carried out. The combination of organic layer was washed with brine, dried over MgSO_4_ and concentrated in vacuo. Purification of the residue by flash chromatography (PE/EtOAc = 6:1) afforded **13** (22.0 mg, 86% yield) as a colorless oil.

#### 4.3.6. Synthesis of Methyl 9-oxononanoate (**27**)

Ozone was bubbled into a solution of **26** (276.4 mg, 0.93 mmol) in CH_2_Cl_2_ at −78 °C until the solution turned to blue and the O_2_ was bubbled into the blue solution at −78 °C until the disappearance of the blue color. PPh_3_ (978.0 mg, 3.7 mmol) was added, and the resulting solution was slowly warmed to 25 °C to keep stirring overnight. The crude mixture was combined with silica gel and concentrated in vacuo. Purification of the residue by flash chromatography (PE/EtOAc = 20:1) afforded **27** (154.3 mg, 89% yield) as a colorless oil.

#### 4.3.7. Synthesis of Methyl (9*E*,11*Z*)-12-bromopentadeca-9,11-dienoate (**14**)

To a solution of **S5** (277.9 mg, 1.03 mmol) in THF (20 mL) at −78 °C was added NaHMDS (0.51 mL, 2 mmol/L in THF). The mixture was stirred at −78 °C for 30 min before **27** (75.6 mg, 0.41 mmol) in THF (5 mL) was added. The mixture was stirred at −78 °C for 2 h before the addition of saturated NH_4_Cl solution (10 mL). Extraction with EtOAc was performed and the combination of organic layer was washed with brine, dried over MgSO_4_ and concentrated in vacuo. Purification of the residue by flash chromatography (PE/EtOAc = 100:1) afforded **14** (43.0 mg, 32% yield).

#### 4.3.8. Synthesis of (9*E*,11*Z*)-12-bromopentadeca-9,11-dienoic Acid (**15**)

To a solution of **14** (43.0 mg, 0.13 mmol) in THF (10 mL) and H_2_O (3 mL) was added LiOH (6.2 mg, 0.26 mmol). The mixture was stirred for 5 h. The pH of the mixture was adjusted to about 5 with HCl (2 mol/L). Then the mixture was extracted with EtOAc and the combination of organic layer was washed with brine, dried over MgSO_4_ and concentrated in vacuo. Purification of the residue by flash chromatography (PE/EtOAc = 6:1) afforded **15** (36.1 mg, 88% yield).

### 4.4. General Method for the Synthesis of Horner Reagent

#### 4.4.1. Synthesis of Horner Reagent for Bromo-Olefination

The bromo-olefination of aldehyde was realized based on the reported strategy [16]. Similarly, the synthesis of (*Z*)-1,3-dibromohex-2-ene and (**S3**), diethyl (*Z*)-(3-bromohex-2-en-1-yl)phosphonate (**S4**) and dimethyl (*Z*)-(3-bromohex-2-en-1-yl)phosphonate (**S5**) were conducted according to the referenced approach [16].

#### 4.4.2. Synthesis of Horner Reagent for Olefination

The synthesis of dimethyl (*E*)-hex-2-en-1-ylphosphonate (**S8**) was commenced with (*E*)-hex-2-en-1-ol (**S6**). (*E*)-hex-2-en-1-ol (**S6**) (1.0 g, 7.5 mmol) was added into CH_2_Cl_2_ and then PBr_3_ (3.1 g, 11.3 mmol) was added slowly at 0 °C. The mixture was stirred for 30 min before adding the saturated NaHCO_3_ solution (10 mL). It was extracted with EtOAc and the combination of organic layer was washed with brine, dried over MgSO_4_ and concentrated in vacuo to obtain the crude (*E*)-1-bromohex-2-ene (**S7**, 875.9 mg). The mixture of **S7** (875.9 mg, 5.4 mmol) and P(OCH_3_)_3_ was heated to 130 °C for 4 h, followed by evaporation for the low boiling point impurities. The mixture was purified by flash chromatography (PE/EtOAc = 4:1) afforded **S8** (761.7 mg, 74% yield).

### 4.5. Spectroscopic Data of Majusculoic Acid and Its Analogs

The NMR, optical rotations (OR), HRESIMS, and infrared (IR) data of majusculoic acid derivatives (**1**, **9**–**15**) were given below. Although the spectral characteristics of other intermediates (**19**–**27**) could be found in the Appendix A.

(**1**) [α${]}_{D}^{25}$ −4.3 (*c* 1.0, MeOH); ^1^H NMR (400 MHz, CDCl_3_) *δ* 6.33 (dd, *J* = 15.0, 9.9 Hz, 1H), 6.23 (d, *J* = 9.9 Hz, 1H), 5.89–5.79 (dt, *J* = 14.4, 7.2 Hz, 1H), 2.45 (t, *J* = 7.4 Hz, 4H), 2.21 (dd, *J* = 14.4 Hz, 6.8 Hz, 2H), 1.70–1.49 (m, 4H), 1.44–1.36 (m, 1H), 1.29 (m, 1H), 0.92 (t, *J* = 7.4 Hz, 3H), 0.56–0.46 (m, 2H), 0.32–0.23 (m, 2H). ^13^C NMR (100 MHz, CDCl_3_) *δ* 179.5, 136.7, 128.1, 127.8, 127.0, 43.6, 34.1, 33.7, 33.0, 29.3, 21.5, 18.5, 18.2, 13.0, 11.9. IR (KBr, cm^−1^) *ν* 2960, 2925, 2855, 1743, 1434, 1171, 986.83; HRMS (ESI, *m*/*z*) calcd for C_15_H_23_BrO_2_ [M + Na]^+^, 337.0774 and 339.0753, found 337.0781 and 339.0762.

(**9**) ^1^H NMR (400 MHz, CDCl_3_) *δ* 6.30 (ddt, *J* = 15.0, 9.9, 1.3 Hz, 1H), 6.20 (d, *J* = 9.9 Hz, 1H), 5.88–5.75 (m, 1H), 3.67 (s, 3H), a2.43 (t, *J* = 7.3 Hz, 2H), 2.38 (t, *J* = 7.5 Hz, 2H), 2.18 (dd, *J* = 14.3, 7.0 Hz, 2H), 1.65–1.55 (m, 4H), 1.55–1.48 (m, 1H), 1.40–1.29 (m, 1H), 0.90 (t, *J* = 7.4 Hz, 3H), 0.52–0.41 (m, 2H), 0.28–0.20 (m, 2H). ^13^C NMR (100 MHz, CDCl_3_) *δ* 174.2, 136.7, 128.1, 127.8, 127.0, 51.4, 43.6, 34.2, 33.7, 33.0, 29.7, 29.6, 21.5, 18.5, 18.3, 13.0, 11.9. IR (KBr, cm^−1^) *ν* 2961, 2927, 2871, 1557, 1742, 1459, 1377, 1261, 1188, 1081, 967, 801; HRMS (ESI, *m*/*z*) calcd for C_16_H_25_BrO_2_ [M + Na]^+^, 351.0930 and 353.0910, found 351.0942 and 353.0925.

(**10**) [α${]}_{D}^{25}$ −1.6 (*c* 1.0, MeOH); ^1^H NMR (400 MHz, CDCl_3_) *δ* 6.02 (dd, *J* = 9.9, 4.0 Hz, 1H), 5.97 (dd, *J* = 10.0, 4.2 Hz, 1H), 5.62–5.48 (m, 2H), 3.66 (s, 3H), 2.38 (t, *J* = 6.5 Hz, 2H), 2.12 (dt, *J* = 14.5, 7.3 Hz, 2H), 2.03 (dt, *J* = 14.4, 7.0 Hz, 2H), 1.61–1.45 (m, 2H), 1.44–1.37 (m, 2H), 1.33–1.26 (m, 2H), 0.89 (t, *J* = 7.3 Hz, 3H), 0.52–0.39 (m, 2H), 0.27–0.17 (m, 2H). ^13^C NMR (100 MHz, CDCl_3_) *δ* 174.2, 132.3, 132.0, 130.5, 130.4, 51.4, 34.7, 34.2, 34.0, 32.6, 29.6, 22.6, 18.5, 18.3, 13.7, 11.8. IR (KBr, cm^−1^) *ν* 2962, 2925, 1710, 1608, 1423, 868; HRMS (ESI, *m*/*z*) calcd for C_16_H_26_O_2_ [M + Na]^+^, 273.1830, found 273.1828.

(**11**) [α${]}_{D}^{25}$ −4.9 (*c* 1.0, MeOH); ^1^H NMR (400 MHz, CDCl_3_) *δ* 6.02 (dd, *J* = 10.1, 4.4 Hz, 1H), 5.98 (dd, *J* = 10.8, 4.8 Hz, 1H), 5.70–5.50 (m, 1H), 2.42 (t, *J* = 7.5 Hz, 2H), 2.12 (dd, *J* = 14.2, 7.1 Hz, 2H), 2.03 (q, *J* = 7.1 Hz, 2H), 1.62–1.44 (m, 2H), 1.47–1.33 (m, 2H), 1.37–1.27 (m, 2H), 0.89 (t, *J* = 7.3 Hz, 3H), 0.47 (m, 2H), 0.24 (m, 2H). ^13^C NMR (100 MHz, CDCl_3_) *δ* 179.7, 132.4, 132.0, 130.5, 130.4(5), 34.7, 34.2, 34.0, 32.6, 29.3, 22.6, 18.6, 18.2, 13.7, 11.8. IR (KBr, cm^−1^) *ν* 2959, 2926, 2858, 2371, 2323, 1710, 1453, 1287, 986; HRMS (ESI, *m*/*z*) calcd for C_15_H_24_O_2_ [M + Na]^+^, 259.1674, found 259.1670.

(**12**) [α${]}_{D}^{25}$ −39.8 (*c* 1.0, MeOH); ^1^H NMR (400 MHz, CDCl_3_) *δ* 6.31 (dd, *J* = 15.1, 9.8 Hz, 1H), 6.20 (d, *J* = 9.9 Hz, 1H), 5.87–5.75 (m, 1H), 4.11 (q, *J* = 7.1 Hz, 2H), 2.43 (t, *J* = 7.2 Hz, 2H), 2.22 (dd, *J* = 14.0, 7.0 Hz, 2H), 1.65–1.56 (m, 2H), 1.43–1.35 (m, 4H), 1.25 (t, *J* = 7.1 Hz, 3H), 1.16 (dt, *J* = 6.7, 4.5 Hz, 1H), 0.90 (t, *J* = 7.4 Hz, 3H), 0.73–0.67 (m, 1H). ^13^C NMR (100 MHz, CDCl_3_) *δ* 174.4, 135.7, 128.5, 127.6, 127.3, 60.4, 43.6, 32.7, 32.5, 22.4, 21.4, 20.3, 15.4, 14.3, 12.9. IR (KBr, cm^−1^) *ν* 3457, 2961, 2931, 2873, 1728, 1682, 1453, 1410, 1178, 1081, 970, 859.

(**13**) [α${]}_{D}^{25}$ −17.5 (*c* 1.0, MeOH); ^1^H NMR (400 MHz, CDCl_3_) *δ* 6.34 (dd, *J* = 15.1, 9.8 Hz, 1H), 6.23 (d, *J* = 9.9 Hz, 1H), 5.90–5.75 (m, 1H), 2.46 (t, *J* = 7.2 Hz, 2H), 2.26 (dd, *J* = 13.7, 6.9 Hz, 2H), 1.62 (dd, *J* = 14.6, 7.3 Hz, 2H), 1.50–1.44 (m, 2H), 1.46–1.38 (m, 1H), 1.28–1.23 (m, 2H), 0.92 (t, *J* = 7.4 Hz, 3H), 0.86–0.78 (m, 1H). ^13^C NMR (100 MHz, CDCl_3_) *δ* 180.0, 135.4, 128.7, 127.6, 127.4, 43.6, 32.7, 32.4, 23.4, 21.4, 20.0, 16.2, 12.9. IR (KBr, cm^−1^) *ν* 2961, 2927, 2854, 1703, 1694, 1454, 1429, 1229, 1080, 968, 878, 802; HRMS (ESI, *m*/*z*) calcd for C_13_H_19_BrO_2_ [M + Na]^+^, 309.0461 and 311.0440, found 309.0468 and 311.0449.

(**14**) ^1^H NMR (400 MHz, CDCl_3_) *δ* 6.31 (dd, *J* = 14.9, 9.8 Hz, 1H), 6.22 (d, *J* = 9.9 Hz, 1H), 5.87–5.74 (m, 1H), 3.69 (s, 3H), 2.45 (t, *J* = 7.2 Hz, 2H), 2.32 (t, *J* = 7.5 Hz, 2H), 2.12 (q, *J* = 7.0 Hz, 2H), 1.67–1.57 (m, 4H), 1.42 (m, 2H), 1.33 (m, 6H), 0.92 (t, *J* = 7.4 Hz, 3H). ^13^C NMR (100 MHz, CDCl_3_) *δ* 174.3, 137.1, 128.0, 127.8, 126.8, 51.5, 43.6, 34.1, 32.9, 29.1, 29.1, 29.0, 24.9, 21.5, 13.0. IR (KBr, cm^−1^) *ν* 3648, 2927, 2854, 1742, 1621, 1260, 1016, 866; HRMS (ESI, *m*/*z*) calcd for C_16_H_27_BrO_2_ [M + Na]^+^, 353.1087 and 355.1066, found 353.1099 and 355.1080.

(**15**) ^1^H NMR (400 MHz, CDCl_3_) *δ* 6.29 (ddt, *J* = 14.8, 9.8, 1.3 Hz, 1H), 6.20 (d, *J* = 9.9 Hz, 1H), 5.85–5.74 (m, 1H), 2.43 (t, *J* = 7.2 Hz, 2H), 2.35 (t, *J* = 7.5 Hz, 2H), 2.10 (q, *J* = 6.9 Hz, 2H), 1.67–1.55 (m, 4H), 1.45–1.33 (m, 4H), 1.35–1.27 (m, 6H), 0.90 (t, *J* = 7.4 Hz, 3H). ^13^C NMR (100 MHz, CDCl_3_) *δ* 179.6, 137.0, 128.0, 127.8, 126.8, 43.6, 34.0, 32.9, 29.1, 29.0(6), 29.0, 24.7, 21.5, 13.0. IR (KBr, cm^−1^) *ν* 2960, 2929, 2855, 1710, 1462, 1464, 1428, 1259, 967; HRMS (ESI, *m*/*z*) calcd for C_15_H_25_BrO_2_ [M + Na]^+^, 339.0930 and 341.0910, found 339.0945 and 341.0924.

### 4.6. Anti-Inflammatory Activity Testing

Nitric oxide (NO) production in mouse macrophages (RAW264.7) was used to detect the inhibitory effect of compounds in inflammatory reaction. When immune cells are stimulated by microbial endotoxin and inflammatory mediators, many induced NO synthase (iNOS) will be generated to produce NO for immune response. Therefore, inhibition of NO production is a direct indicator of anti-inflammatory activity of compounds. The inhibitory activity of majusculoic acid and its analogs on NO production in LPS-induced inflammatory models were evaluated in our research.

Griss kit was used to detect NO content in the culture medium (Molecular probes, G-7921). Macrophages (RAW264.7) were cultured in Dulbecco modified eagle’s medium (DMEM) medium which contained 10% fetal bovine serum (FBS). The cell concentration of macrophages was adjusted and then inoculated in a 24-well cell culture plate for 24 h. The test compounds of different concentrations were pretreated for 2 h before the incubating of LPS (100 ng/mL) for 24 h. The blank medium was used as control. After adding lipopolysaccharide (LPS, 1 µg/mL) for 24 h, the supernatant culture medium was combined with Griss reagent, and the absorbance value was measured at 548 nm by spectrometry.
NO production inhibition rate = (OD_control_ − OD_sample_)/(OD_control_ − OD_blank_) × 100%,(1)

### 4.7. Cytotoxicity Determination

It is expected that compounds inhibited the LPS-induced inflammatory response should not have cytotoxicity towards cells. Therefore, all eight compounds were also tested for cell proliferation by CCK-8 assay. Briefly, RAW264.7 cells were inoculated in 96-well plates and cultured for 24 h. The tested compounds were added, and incubation was continued for another 72 h. Then 10 μL CCK-8 solution was add to each well. After 1.5 h, the OD values at 450 nm were determined.
Cell proliferation inhibition rate (%) = (OD_control_ − OD_sample_)/(OD_control_ − OD_blank_) × 100%,(2)

## 5. Conclusions

Our work provides a novel and efficient strategy to realize the total synthesis of natural majusculoic acid and other seven derivatives in ten steps with yield of 3% to 28%. Three compounds, majusculoic acid (**1**), methyl majusculoate (**9**), and ethyl (1*R*,2*R*)-2-((3*E*,5*Z*)-6-bromonona-3,5-dien-1-yl) cyclopropane-1-carboxylate (**12**), exhibited weak anti-inflammatory effects with inhibition rates of 33.68%, 35.75%, and 43.01% in vitro. This work could give assistance to more in-depth structure–activity relationship (SAR) research of CFAs.

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
