# Peer review of "Total Synthesis and Anti-Inflammatory Bioactivity of (−)-Majusculoic Acid and Its Derivatives"

_marinedrugs, 2021, doi:10.3390/md19060288_

Round 1
Reviewer 1 Report
The authors described the total synthesis of Majusculoic Acid and its derivatives. This is not the first total synthesis of the Majusculoic Acid but their synthesis is more concise and allowed them to synthesize several derivatives. Here are few improvement suggestions:
1- There are general English issues; some sentences needs to be revised. Few examples but there are others:
To find out novel anti-inflammatory compounds from CFAs is possible.
In order to know better about the structure-activity relationship, together with the development of the other strategies to obtain the designed majusculoic acid derivatives.
Through this way, it is successful to construct the trans-cyclopropane unit, extending the carbon chain in those 83 eight steps.
2- Compounds 12 and 23 spectra shows significant amount of impurity.
Reviewer 2 Report
This manuscript describes synthesis of marine natural product jusculoic acid and its derivatives and evolution of them for anti-inflammatory activities as well as anti-proliferative effects. Although the body text of this manuscript may reach the standard of this journal, it unfortunately lacks compound characterization of synthetic intermediates in the experimental part. Compound characterization is very important and required not only for the final products such as compound 1, 9, etc but also for synthetic intermediates such as compounds 18, 19, etc. Additionally, structural forms in Figure 2 should have R groups (for acids and esters). Accordingly, the reviewer thinks that this manuscript should be accepted after revision including characterization for all compounds.
Round 2
Reviewer 2 Report
The reviewer checked the revised manuscript and the supporting information, and now agrees that the manuscript is qualified enough to be published.